# Environmental Flow Releases for Wetland Biodiversity Conservation in the Amur River Basin

**Oxana I. Nikitina** [1],[*] ⓘ**, Valentina G. Dubinina** [2]**, Mikhail V. Bolgov** [3]**, Mikhail P. Parilov** [4] **and Tatyana A. Parilova** [4]

1   World Wide Fund for Nature (WWF-Russia), Moscow 109240, Russia
2   Central Directorate for Fisheries Expertise and Standards for the Conservation, Reproduction of Aquatic Biological Resources and Acclimatization, Moscow 125009, Russia; vgdu@mail.ru
3   Water Problems Institute of the Russian Academy of Sciences, Moscow 117971, Russia; bolgovmv@mail.ru
4   Khingan Nature Reserve, Arkhara 676748, Russia; mparilov@mail.ru (M.P.P.); tkuznetsova@mail.ru (T.A.P.)
*   Correspondence: onikitina@wwf.ru; Tel.: +7-910-462-90-57

**Abstract:** Flow regulation by large dams has transformed the freshwater and floodplain ecosystems of the Middle Amur River basin in Northeast Asia, and negatively impacted the biodiversity and fisheries. This study aimed to develop environmental flow recommendations for the Zeya and Bureya rivers based on past flow rate records. The recommended floodplain inundation by environmental flow releases from the Zeya reservoir are currently impracticable due to technical reasons. Therefore, the importance of preserving the free-flowing tributaries of the Zeya River increases. Future technical improvements for implementing environmental flow releases at the Zeya dam would improve dam management regulation during large floods. The recommendations developed for environmental flow releases from reservoirs on the Bureya River should help to preserve the important Ramsar wetlands which provide habitats for endangered bird species while avoiding flooding of settlements. The results emphasize the importance of considering environmental flow during the early stages of dam planning and the need to enhance the role of environmental flow in water management planning.

**Keywords:** Amur; hydropower; dam; damage; environmental flow releases; biodiversity conservation; freshwater ecosystems; wetlands

## 1. Introduction

### 1.1. The Urgent Need for Restoring Freshwater Biodiversity

In recent decades, the biodiversity of freshwater ecosystems has been sharply decreasing. The 2020 Living Planet Index shows that the average abundance of freshwater populations monitored across the globe has declined by 84% since 1970 [1], and the extinction level of freshwater fish in the 20th century was the highest in the world among vertebrates [2]. The operation of dams has led to changes in flow regimes of many rivers in the world [3–6] including in Russia [7–11], and has depleted and changed the species composition of rivers, which makes freshwater ecosystems unstable to the effects of natural and anthropogenic factors. Irreversible water withdrawal and water pollution, water operation without effective means of protection of freshwater ecosystems, irrigation, and poaching have served as additional factors contributing to a sharp decline in freshwater biodiversity [12]. Urgent measures should be adopted to address freshwater biodiversity loss [13]. Studies have demonstrated that changes in freshwater ecosystems could be reversible and that ecosystems could retain sufficient potential for recovery [14]. The conservation and restoration of freshwater ecosystems should be a key component of sustainable water resources management, while biological productivity should be

used as an indicator of the state of freshwater ecosystems [15]. A key element in the restoration of disturbed freshwater ecosystems is ensuring the hydrological regime for favorable environmental conditions within freshwater and floodplain ecosystems. The science and practice of environmental flow assessment enables identification and quantification of these attributes [1,16,17].

### 1.2. Environmental Value of the Amur River Basin

The Amur River, in Northeast Asia, is one of the ten largest rivers in the world. Its basin covers over 1.8 million square kilometers of Russian, Chinese, and Mongolian land [18] (see Figure 1). The state Sino-Russian border along the Amur River and its tributaries reaches 3500 km.

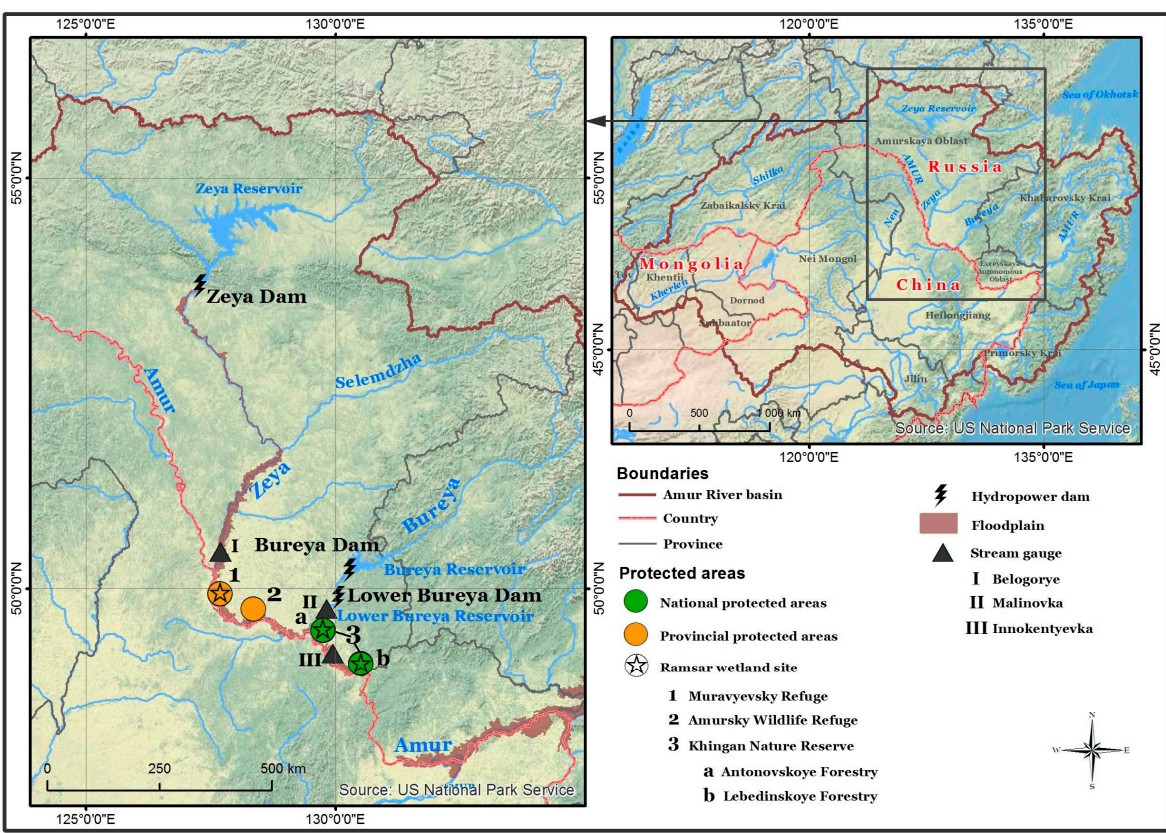

**Figure 1.** Large dams in the Russian part of the Amur River basin and the important wetlands, including Ramsar sites, located within protected areas under the dam impact.

The Amur River basin ecosystems maintain high levels of biodiversity [19], providing habitats for 130 fish species, 18 of which are endemic. The largest salmon and sturgeon populations of the Pacific Ocean live in the Amur River basin [20,21]. Endangered birds such as Oriental storks (*Ciconia boyciana*), red-crowned cranes (*Grus japonensis*), and white-naped cranes (*Grus vipio*) breed in the Amur wetlands. There are 320 terrestrial vertebrate fauna species that inhabit the floodplains of the Amur River. Flora assemblage of the riverside forests is estimated at more than 300 vascular plant species [22]. Despite the regulated flow of the main tributaries, the main channel of the Amur River remains free of dams, providing ecological integrity to the basin [23].

### 1.3. Amur River Basin and Floods

#### 1.3.1. Natural Features

The river flow in the Amur River basin is influenced by rainfall, reaching 80% of the annual flow [24]. Frequent floods which are caused by the monsoons of Eastern Asia are common [25,26].

During the warm season from May to September, four to five floods occur, which have caused water level fluctuations of up to 6–8 m in large rivers. The maximum water levels are typical for July and August. Seasonal flow fluctuations in the Amur River basin are exceptionally large as compared with other regions of Russia.

Floods are formed within six major tributaries of the Amur River catchment, i.e., the Argun, Shilka, Zeya, Bureya, Sungari, and Ussuri rivers. Flood magnitude is determined by the maximum flow rates and flood levels on the tributaries as well as by flood volume [25,27]. Floods formed in a separate catchment area can also lead to a large flood on the Amur itself. Significant rises in water levels occur once every two to three years. Once every 10–15 years, floods on several large tributaries coincide, which lead to inundation of the Amur River floodplain and the adjacent areas [27].

### 1.3.2. The Role of Water Flow and Floods in Freshwater and Floodplain Ecosystem Maintenance and Function

Natural water flow provides ecosystem services such as maintaining specific hydraulic and geomorphological stream parameters (non-silting velocity, physical and chemical properties of water, etc.), and maintaining the area, timing, and duration of flooding of a floodplain. River flow which defines the thermal regime, water turbidity, as well as soil and vegetation cover, plays a vital role within freshwater and floodplain ecosystems. The permissible recurrence of ecologically unfavorable hydrological conditions should be determined by the characteristics of freshwater ecosystems and their inertia [15,28].

During high floods, the Amur River overflows its riverbanks for tens of kilometers and forms a backwater with its tributaries, bypassing their floodplain ecosystems and washing out oxbow lakes. Floods have a positive effect on fish reproduction, create favorable conditions for the migration of anadromous fish, and enhance the fertility of floodplain soils, creating a high potential for floodplain biological productivity [22,29]. Therefore, floods contribute to freshwater and floodplain ecosystem maintenance in the Amur River basin.

### 1.3.3. Socioeconomic Reasoning for Dam Building

Floods are the main natural disaster in the Amur River basin, negatively affecting the region's economy [30–33]. The most significant losses are common in the agriculture, industrial, and communal sectors [30,33].

Large dams regulate the flow of the following three main Amur tributaries: the Zeya and Bureya rivers in Russia, and the Songhua River in China. In Russia, three large dams (see Figure 1) have been built to produce electricity and protect areas from floods [25]. In the Zeya River basin, the Zeya dam (1330 MWt), located 640 km from the Zeya mouth, has been operating since 1984 [34]. The dam regulates 45% of the total Zeya flow [35]. On the Bureya River, the Bureya dam (2010 MWt), located 174 km from the Bureya mouth [36], has been operating since 2003. Since 2017, the Lower Bureya dam (320 MWt), located 85 km from the Bureya mouth [37] has also been operational. In general, the reservoirs successfully protect the human population from floods [25].

However, the catastrophic flood of 2013, which lasted over two months, became the largest flood in the Amur River basin for the last hundred years [24,38–40] and demonstrated the limitations of dams to protect the population. In Russia, tens of thousands of people were evacuated and many lost homes. In China, the flood also brought great disasters, including human casualties [33]. After this extraordinary flood, plans to build from four to ten new dams for additional flow regulation in Russia were announced [32,41,42], but have not yet been implemented.

*1.4. Impact of Dams on the Regulated Zeya and Bureya Ecosystems*

1.4.1. Long-Term Flow Regulation and Its Impact on Freshwater Ecosystems of the Zeya and Amur Rivers

As a result of flow regulation, maximum water flow rates of the Zeya River have decreased by more than 20%, and the frequency of floods has declined [43], while, as a result of irregular flooding, the settlement area within the Zeya floodplain has increased up to 1.5 times [44]. After construction of the Zeya dam, the Amur River water flows during summer reduced [45,46], which led to a deterioration of water exchange in floodplain wetlands and a gradual overgrowth of oxbow lakes [47,48]. The wetlands of the Muravyevsky Refuge (Ramsar Site, Wetland of International Importance) and the Amur Reserve in the Amur floodplain are affected by the flow regimes of the Amur and Zeya rivers. These wetlands are the habitat of endangered bird species such as the Oriental stork (*Ciconia boyciana*), red-crowned crane (*Grus japonensis*), white-naped crane (*Grus vipio*), and hooded crane (*Grus monacha*) [49].

Construction of the Zeya hydropower dam is an obstacle for fish migrations. The species composition of the Zeya reservoir, with an area of 2500 km$^2$, has decreased by a third; 12 out of 38 fish species have disappeared from the waterbody, including endangered species such as the kaluga (*Huso dauricus*) and Amur sturgeon (*Asipenser schrenckii*), while two invasive species have appeared (*Coregonus peled* and *Coregonus migratorius*) [50]. Species depletion and composition change make freshwater ecosystems unstable to the effects of natural and anthropogenic factors [28,51]. The Zeya reservoir was originally planned as a fishery reservoir; the main commercial catch was the Amur pike (*Esox reichertii*). However, the current pike catches are five to seven tons versus the expected 400 tons per year [52].

Fish stocks have decreased in all the water bodies located within the impacted area of the dam. In the 1980s, commercial fish productivity of the Zeya River was 20–25 kg per ha and the productivity of the floodplain lakes was 30–40 kg per ha. By 2008, the number had decreased to 0.34 and 0.22 kg per ha [53]. Before the dam construction, the main species of the Zeya river were fish laying eggs in the inundated terrestrial vegetation, i.e., phytophilic fish, representing up to 70–80% of the total freshwater fish stocks, including Amur catfish (*Silurus asotus*), Amur pike (*Esox reichertii*), the Prussian carp (*Carassius gibelio*), and the Amur carp (*Cyprinus carpio*). Due to decreased summer flows, the phytophilic fish spawning ground range and food supply has decreased, which has led to a decrease in their numbers [54]. The influence of cold water is found in the Zeya River section from the dam up to the confluence of the large free-flowing Selemdzha River, which has also changed the species composition towards lithophilic fish species [50,55].

Conversely, the maximum winter discharges and the water levels of the Middle and Lower Amur River basin have significantly increased [45,46]. This has led to intensified formation of intra-water ice, sludge, and congestion, which could threaten the stable operation of water intakes and other economic facilities [56]. Although an increase in the winter flow prevents fish suffocation, it does not contribute to a rise in their number [54].

1.4.2. Effects of Flow Regulation on Freshwater Ecosystems of the Bureya River Basin

The Khingano-Arkharinskaya Lowland (Ramsar Site, Wetland of International Importance) is impacted by the dams on the Bureya River (see Figure 1). The area provides breeding habitats for endangered bird species such as the Oriental stork (*Ciconia boyciana*), red-crowned crane (*Grus japonensis*), and white-naped crane (*Grus vipio*) (see Figure 2). Up to 10% of the world's Oriental stork population breeds here, and a large group of red-crowned cranes inhabit this area [57]. One of the main conditions for bird nesting is the location of nests in close proximity to weak-flowing water bodies inhabited by small fish [58,59]. The reduced fish abundance in lakes has worsened the living conditions for birds and lead to the loss of nesting area and a decrease in the number of nesting pairs [60].

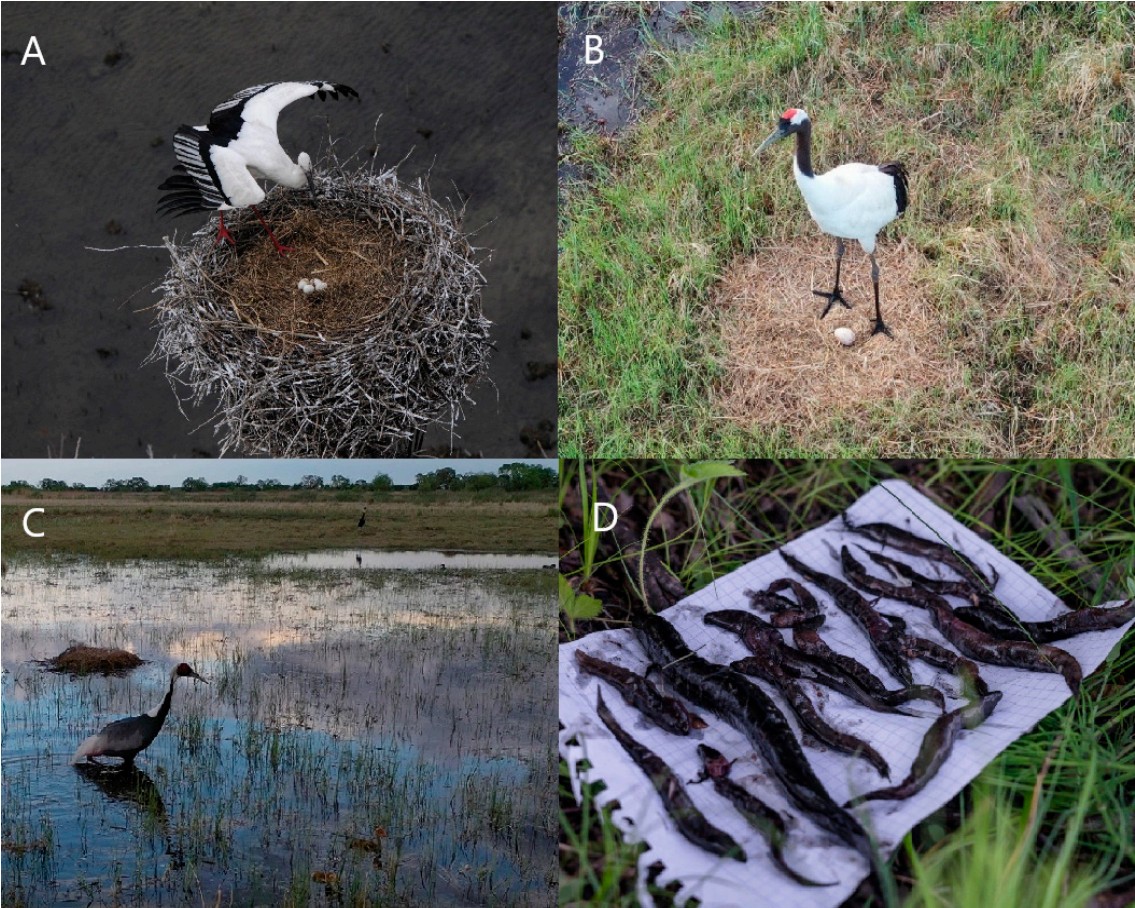

**Figure 2.** The endangered bird species inhabiting wetlands under the impact of dams, and their food supply. (**A**) Oriental stork (*Ciconia boyciana*), by D. Korobov; (**B**) Red-crowned crane (*Grus japonensis*), by A. Sasin; (**C**) White-naped crane (*Grus vipio*), by A. Sasin; (**D**) Loaches (*Misgurnus mohoity*) and Amur sleeper (*Perccottus glenii*), typical food for storks and cranes, by E. Egidarev.

After the Bureya dam started operating in 2002, the magnitude of the high floods was reduced. The reduction in maximum water levels caused limited and infrequent water flow along small river channels and lakes, which resulted in their gradual overgrowth. The negative effect of the dam's operation has been reinforced during dry periods and has led to the loss of nesting sites for cranes [61]. At the peak of the drought in 2001–2003, the number of breeding cranes and storks decreased to a minimum. Under natural flow conditions, high floods have compensated for the adverse effects of droughts, restoring the productivity of wetlands [60].

Before dams were built, 36 fish species inhabited the Bureya River basin. After dam construction, the number of fish species in the Bureya reservoir decreased to 27 species of freshwater fish, while the number of fish species downstream from the dams decreased to 20 species [62].

*1.5. Optimizing Reservoir Operating Rules to Meet Environmental Flows and Minimize the Ecological Effects of Flow Regulation*

In order to reduce the negative impact of flow regulation and preserve freshwater and floodplain ecosystems, the flow regime in the warm season should mimic the natural flow by implementing environmental flow releases from reservoirs. These environmental flow releases should maintain river channels and spawning grounds which favorably affect the environment for fish, reconnect oxbow lakes and maintain a water regime of wetlands close to natural, and increase soil fertility within floodplains.

Russian official agencies responsible for assessing and limiting human impact on water bodies follow the Water Code when developing Standards for Permitted Limits of Impact on Water Bodies

and Comprehensive Schemes for Water Bodies Management and Protection [63]. During the scheme's development, environmental flows are established for multiple river stretches in order to assess the water balance of the river basins, which are required in order to calculate the permitted water withdrawal for a particular water management area during the development of standards. In 2014, both the standards and scheme were developed for the Amur River basin and approved by the official agencies for the next decade [64,65]. However, these documents do not include sufficient requirements for maintaining the sustainable state of the freshwater and floodplain ecosystems [66]. Furthermore, in order to implement environmental flow releases from the reservoirs, such requirements ought to be outlined in the Reservoir Operating Rules, the underlying document on operational reservoir management [67].

Currently, environmental flow releases from the reservoirs on the Zeya and Bureya rivers have not been implemented, except for sanitary flows, to ensure compliance with water quality standards downstream from the dams. These sanitary flows are not enough to maintain freshwater or floodplain ecosystems. No previous recommendations for the implementation of environmental flow releases with broader objectives have been developed, although the nature conservation community and environmental experts have repeatedly pointed out their importance in resolving this issue [47,68–70].

Alongside the construction of the Lower Bureya Dam, an environmental compensation program was undertaken by the hydropower company RusHydro, the local government, the UNDP/GEF, and ecologists [70,71]. The project aimed to minimize impacts of the construction of the Lower Bureya dam on the biodiversity of the terrestrial ecosystem but did not include measures such as environmental flow releases aimed at conserving freshwater ecosystems or species [70,72].

## 2. Materials and Methods

### 2.1. Materials

The environmental flow requirements for the Zeya River basin were estimated according to the methodological approaches for determining surface water withdrawal and environmental flow (release) [73,74], which is the main approach applied in Russia for assessing environmental flow. For the calculations, the data on daily mean discharge for 1957–2017 at the Belogorye stream gauge was used. The stream gauge is located in the lower reaches of the Zeya River, i.e., 43 km from the river mouth, and 617 km from the dam (see Figure 1). The natural flow regime was analyzed using the data from 1957 to 1974, the period before the dam started operating.

To assess the environmental flow releases from the Bureya reservoirs, floods under natural and regulated flow regime were analyzed. The data on daily mean discharge from 1957 to 2017 at the Malinovka stream gauge, located 80 km from the mouth of the Bureya, downstream of the Bureya and Lower Bureya dams (see Figure 1), were used. The natural flow regime was analyzed using the data from 1957 to 1999, as the Bureya River was blocked for dam construction in 2000.

The flow rates for the water overflow from the channel into the adjacent floodplain were also determined from the Belogorye and Malinovka stream gauges.

### 2.2. Methodology for Assessment of Permitted Surface Water Withdrawal and Environmental Flow (Release)

The method is based on the principle of sustainable functioning of freshwater and floodplain ecosystems and preservation of natural breeding conditions.

The methodological approaches for determining the permitted surface water withdrawal and assessing environmental flow (release) are based on published materials [28,51,73,74]. The components of ecosystems in river basins depend on the ecologically significant elements of the hydrological regime that characterize their state. For rivers, an ecologically significant element of the hydrological regime is flow velocity and flow discharge. The volume of runoff describing the optimal and normal conditions should be determined, as well as water volume for critical conditions, when a sharp deterioration in living conditions and only minimal natural reproductive processes occur.

The river discharge which corresponds to the overflow of water from the channel into the floodplain can be used as a basis for establishing critical conditions [15,28,73,75]. The critical average daily water discharge was determined according to the hydrological data, as well as the corresponding discharge values when the water does not overflow into the floodplain. Then, the corresponding value of the critical annual runoff was determined.

The permitted surface water withdrawal is the maximum volume of water withdrawn from the river basin, which preserves the conditions for the stable and safe functioning of freshwater and floodplain ecosystems or their components. The value of the permitted surface water withdrawal from the river should preserve the intra-annual flow fluctuations, as close as possible to natural conditions without exceeding the limits of natural long-term fluctuations. In order to define the permitted surface water withdrawal, ecological criteria such as the conditions of natural reproductive processes of aquatic biological resources, the structure of the fish community, and the species diversity are used.

Environmental flow is calculated as the difference between the runoff volume and its permitted withdrawal. Therefore, this is the runoff with an admissible irreversible water withdrawal, which provides conditions for stable and safe ecosystem function and restoration.

The algorithm for defining the environmental flow is as follows:

1.  On the basis of an analysis of the relationship among natural hydrological characteristics and freshwater ecosystem productivity or other indirect indicators, the critical discharge and volume of water, $Q_{cr}$ and $W_{cr}$ are defined, corresponding to the critical state of freshwater ecosystem.
2.  The historically minimal runoff volume ($W_{hist}$) is taken as the restored minimum runoff per year with 99% probability of exceedance.
3.  The difference between the critical runoff volume and the historically minimal runoff determines the volume which can be withdrawn from the river with minimal damage to the ecosystem in the long term. The average volume of permissible irrevocable withdrawal is determined by the formula:

$$W_{iw\ mean} = W_{cr} - W_{hist} \tag{1}$$

4.  The intra-annual distribution of $W_{iw\ mean}$ is carried out according to the long-term natural flow regime.
5.  The permitted water withdrawal is determined for the stream gauge located in the lower reaches of the river basin, mostly close to the river mouth. The withdrawal volume is determined for different water year types (e.g., normal, very wet, wet, dry, and very dry), considering intra-annual seasons of the water year:

$$W_{iw\ j} = W_{iw\ mean} \times (W_j/W_{mean}) \tag{2}$$

where $W_j$ is the natural (or restored) runoff in the year with j% probability of exceedance and $W_{mean}$ is the mean natural (restored) runoff in the lower reaches of the river basin.

6.  The permitted water withdrawal value for the river sections located upstream is defined along the main river channel according to the formula:

$$W^i_{iw\ j} = K \times W_{iw\ j} \tag{3}$$

where $W_{iw\ j}$ is the permitted water withdrawal volume established for the whole basin for the year of j% probability of exceedance and K is the proportion between the total river runoff for the year with j probability of exceedance and the river runoff in a defined river section.

7.  In dry years with a runoff below the critical volume, water withdrawal is allowed only for priority needs, such as drinking and domestic water supply.

8.　　According to the defined value of the permitted water withdrawal, the environmental flow (WiEFlow j) is calculated for the year of j% probability of exceedance:

$$W^i_{EFlow\,j} = W^i_j - W^i_{iw\,j} \tag{4}$$

9.　　The environmental flow release from the reservoir should consider the needs for fishery and ecosystem maintenance, channel-forming processes, and ensure proper sanitary requirements. If there is a lateral inflow downstream from the dam, the environmental flow release must ensure compliance with the defined requirements of environmental flow. When establishing recommendations for environmental flow releases, fishery needs can be taken as a basis, providing conditions for natural reproductive processes of commercially valuable and other fish species, as well as other aquatic flora and fauna inhabiting freshwater ecosystems downstream of the dam.

10.　The intra-annual distribution of critical flow, as well as environmental flow and releases is carried out under the intra-annual flow distribution for a particular year.

When determining the permissible irrevocable water withdrawal, first, the calculation is carried out for the entire basin, and then for the upstream sections.

The permitted water withdrawal should not exceed 20% of the average annual runoff volume [76].

## 3. Results

### 3.1. Environmental Flow Assessment in the Zeya River Basin

On the Zeya River section from the hydropower dam up to the confluence of the Zeya and its tributary Selemdzha, the river course is relatively straight, with a small number of tributaries. Such a morphological structure determines the poverty of fish species. The negative effect of cold water on aquatic organisms is found here [55] and is an additional limiting factor in the reproduction of aquatic biological resources. After the confluence of the Zeya and Selemdzha rivers, the Zeya water is sufficiently heated. Its river course is meandering, forming channels, oxbow lakes, and grounds favorable for fish reproduction. A large number of channels and lakes are located within the wide floodplain on the right riverbank. Due to the thermal regime, this river section is comfortable for most freshwater fish living in the Zeya River basin. The Zeya's most important area for fish habitat is the channel and floodplain of the Zeya River below the confluence of the Selemdzha River and up to the Zeya River's lower reaches with a straight river course flowing into the Amur River. The river section between the confluence of the Selemdzha and Zeya rivers is the most ecologically valuable site of the Zeya River basin. The hydrological conditions of this area can be characterized by the data of the Belogorye stream gauge.

To define the environmental flow requirements, floods in the Lower Zeya River basin under the natural flow regime and the flow regulation were compared.

In order to inundate the Zeya floodplain and ensure conditions for natural reproduction processes for phytophilic fish, the river discharge at the Belogorye stream gauge should exceed 6500 m³/s, which corresponds to the discharge of the overbank flow. According to hydrological observations under a natural flow regime between 1956 and 1974, the floodplain inundation at the Belogorye stream gauge has occurred annually and throughout the entire warm season. From early May to late July, during fish spawning, the duration of continuous periods of the floodplain inundation was 15–20 days, reaching 30–37 days some years (see Figure 3a–e). During August and September, at the end of the spawning and fattening periods, the floodplain was also annually inundated (see Figure 3f–h) for 5–32 days, causing water inflow into oxbow lakes. When river discharges exceeded 9000 m³/s, the high rates would wash organic debris and macrophytes out of the lakes.

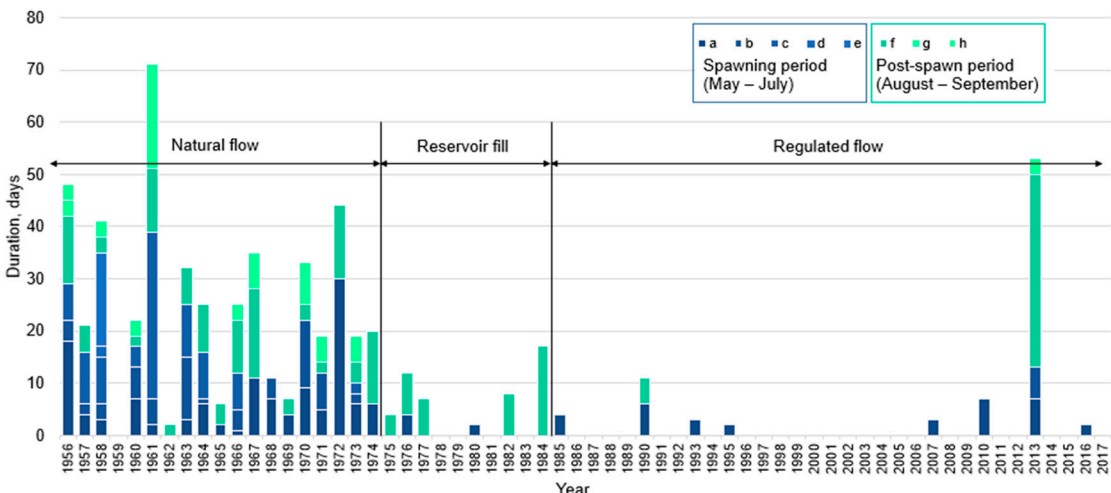

**Figure 3.** Under flow regulation, the frequency and duration of the Zeya River's bank overflow and floodplain inundation at the Belogorye stream gauge decreased, which negatively affects the reproduction of phytophilic fish.

To determine the environmental flow for the Zeya River at the Belogorye stream gauge, critical conditions for the freshwater ecosystem were determined, such as the flow rate and water volume which did not lead to bank overflow. The only relatively dry year in the short natural flow regime observation period of 1957–1974 was 1962 (78% probability of exceedance). Due to a short period of observations, the use of the 1962 data leads to an overestimation of the permitted water withdrawal, which is 20% more than the average flow rate. Such a large water withdrawal cannot contribute to the freshwater ecosystem's preservation. Therefore, the volume of critical flow ($W_{cr}$) is assumed to be equal to the runoff volume for a year with 90% probability of exceedance and equals 45 km$^3$. The difference between the volume of critical flow ($W_{cr}$) and the historically minimal runoff with 99% probability of exceedance ($W_{hist}$) is the average value of the permitted water withdrawal ($W_{iw\ mean}$):

$$W_{iw\ mean} = W_{cr} - W_{hist} = 45 - 35 = 10\ km^3. \tag{5}$$

Therefore, the average permitted water withdrawal is 10 km$^3$, equivalent to 17.5% of the Zeya's 57 km$^3$ average flow. The flow volumes and permitted water withdrawal were determined for wet (25% probability of exceedance), normal (50% probability of exceedance), dry (75% probability of exceedance), and very dry (95% probability of exceedance) water years.

For different water years, the environmental flow volumes are the following: 55 km$^3$ (P = 25%), 48 km$^3$ (P = 50%), 42 km$^3$ (P = 75%), 34 km$^3$ (P = 95%). The environmental flow volume is distributed between the warm and cold seasons in the ratios of 85–90% and 10–15% (see Table 1).

### 3.2. Flood Analysis in the Bureya River Basin for Environmental Flow Determination

The current surface water withdrawal from the Bureya River is 0.008% of the permitted withdrawal value for a very dry year [77] and the problem of excessive water withdrawal or the lack of water resources is not relevant at the present time or in the foreseeable future. Therefore, the volumes of irrevocable permitted water withdrawal have not been established.

We believe that flow regulation should also provide conditions for the preservation of freshwater ecosystems downstream from the dam, and the flows for periodic floodplain inundation should be maintained. To estimate the required river discharges, floods under natural flow regimes and under flow regulations were compared, and recommendations for the environmental flow releases were determined.

**Table 1.** Annual and intra-annual distribution of surface water withdrawal and environmental flow volumes of the Zeya River basin for different water years.

| Characteristic | Annual Volume of Water km$^3$ | Warm Season (May–September) | | Cold Season (October–April) | |
|---|---|---|---|---|---|
| | | Water Volume km$^3$ | Proportion of Annual Flow % | Water Volume km$^3$ | Proportion of Annual Flow % |
| $W_{cr}$ | 45.0 | - | - | - | - |
| $W_{hist}$ | 35.0 | - | - | - | - |
| $W_{iw\ mean}$ | 10.0 | - | - | - | - |
| $W_{25\%}$ (wet year) | 67.0 | 59.0 | 88 | 8.0 | 12 |
| $W_{50\%}$ (normal year) | 58.0 | 52.5 | 91 | 5.5 | 9 |
| $W_{75\%}$ (dry year) | 51.0 | 45.0 | 88 | 6.0 | 12 |
| $W_{95\%}$ (very dry year) | 41.0 | 34.5 | 84 | 6.5 | 16 |
| $W_{iw\ 25\%}$ (wet year) | 12.0 | 10.6 | 88 | 1.4 | 12 |
| $W_{iw\ 50\%}$ (normal year) | 10.0 | 9.1 | 91 | 0.9 | 9 |
| $W_{iw\ 75\%}$ (dry year) | 9.0 | 7.9 | 88 | 1.1 | 12 |
| $W_{iw\ 95\%}$ (very dry year) | 7.0 | 5.9 | 84 | 1.1 | 16 |
| $W_{EFlow\ 25\%}$ (wet year) | 55.0 | 48.4 | 88 | 6.6 | 12 |
| $W_{EFlow\ 50\%}$ (normal year) | 48.0 | 43.4 | 90 | 4.6 | 10 |
| $W_{EFlow\ 75\%}$ (dry year) | 42.0 | 37.1 | 88 | 4.9 | 12 |
| $W_{EFlow\ 95\%}$ (very dry year) | 34.0 | 28.6 | 84 | 5.4 | 16 |

### 3.2.1. Floods under Natural Flow Conditions

Under a natural flow regime, up to five to seven floods were observed during summer and early autumn, and the most significant level rises occurred during July and August. Large floods were typical for the lower reaches of the Bureya River, when the water level rise could reach 6–10 m above the pre-flood water levels, causing flooding of settlements and agricultural lands. Over the 48-year observation period at the time of 1966, the water level fluctuations measured at the Kamenka (Malinovka) stream gauge were 870 cm for the year with a 1% probability and 530 cm for the year with 50% probability of exceedance. Large floods reoccurred every 10–11 years [18]. High floods were observed in 1917, 1961, 1971, 1972, 1975, 1976, and 1984 [78].

The flushing water regime, during natural conditions, prevented the rapid overgrowth of floodplain lakes. The waters of the Bureya River entered the oxbow lakes and flowed south through them into the Amur River [61].

The water level rise, during the extraordinary flood of 1972, amounted to almost 9 m. The maximum discharge reached 17,700 m$^3$/s in July flow and rates above 10,000 m$^3$/s were observed for five days.

The last major flood under natural flow conditions occurred in August 1984. The high level of the Amur River created a backwater to the high flood on the Bureya River, thus the floodplains of the Amur, Bureya and Yarchikha rivers were flooded. The water levels of the floodplain lakes were very high, and the waters of the Bureya River merged with the waters of some oxbow lakes [79]. The flood of 1984 affected the high floodplain of the Bureya River and contributed to an increase in the

fish productivity of its lakes after the drought, which reached its peak from 1980 to 1981. Maximum river discharges of the Bureya river at Kamenka (Malinovka stream gauge) reached 5270–7630 m$^3$/s (with equivalent water levels 529–560 cm above the zero gauge). Local precipitation was 170 mm in August with a climate average of 143 mm [80]. The watering of lakes in the Bureya floodplain occurred under the following simultaneous conditions:

1. High water level of the Amur River;
2. High water level (discharge) of the Bureya River;
3. Prolonged and abundant local precipitation.

During the flood, the lakes were washed out of macrophytes. It can be assumed that the washing of the lakes was ensured by the passage of a flood wave with a flow rate of more than 7000 m$^3$/s in combination with the high level of the Amur River and an excessive amount of precipitation. Thus, the probable value of the flushing flow rate for the Bureya river at the Malinovka stream gauge is 7000 m$^3$/s.

The years after the flood of 1984 were the most productive in terms of the number of breeding pairs of cranes and storks for all the observed years [60].

### 3.2.2. Floods under the Flow Regulation

Since the beginning of the Bureya dam operation in 2003, the maximum flow rate during summer floods has significantly reduced (see Figure 4).

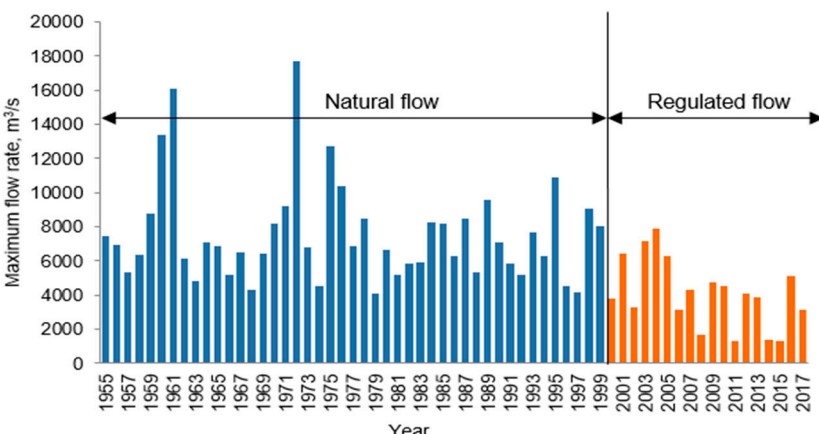

**Figure 4.** Maximum levels of the Bureya River at the Malinovka stream gauge decreased due to flow regulation by dams.

The summer flood of 2013 was not extraordinary in the Bureya River basin, yet the water levels were high as compared with an average summer. The maximum inflow to the Bureya reservoir reached 5175 m$^3$/s, while the maximum water release from the reservoir was 3670 m$^3$/s (see Figure 5), and the maximum flow rate at the Malinovka stream gauge reached 3800 m$^3$/s [81]. The provision of the maximum daily inflow is estimated at approximately 80% probability of exceedance; as a result of flow regulation, the maximum water level decreased at the Malinovka stream gauge by about 0.5 m [24].

The high-water level of the Amur River caused the backwater of the Bureya River and led to water overflow into the floodplain and hydrological connectivity of the Bureya River and the floodplain wetlands. Water of the Yarchikha River began to enter the Dolgoye Lake on August 19 (see Figure 6). On that day, the water level of the Bureya River at the Malinovka stream gauge was 395 cm, the river discharge was 3690 m$^3$/s, and the water release from the Bureya reservoir was 3670 m$^3$/s [81].

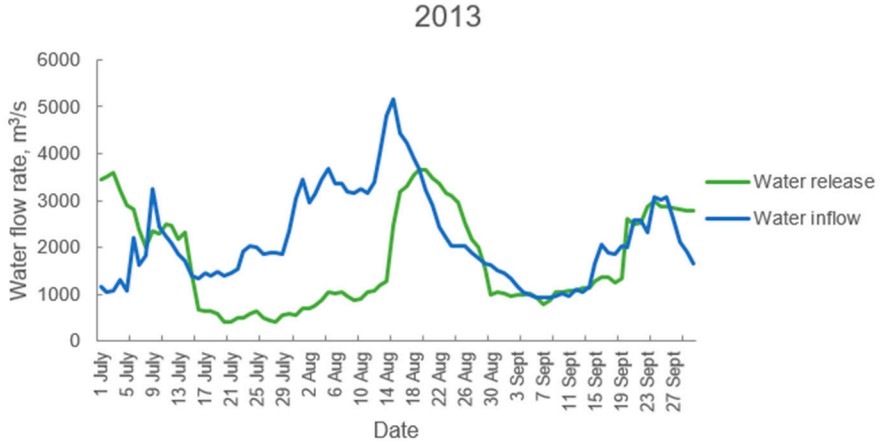

**Figure 5.** Water inflow to and releases from the Bureya reservoir during the flood of 2013.

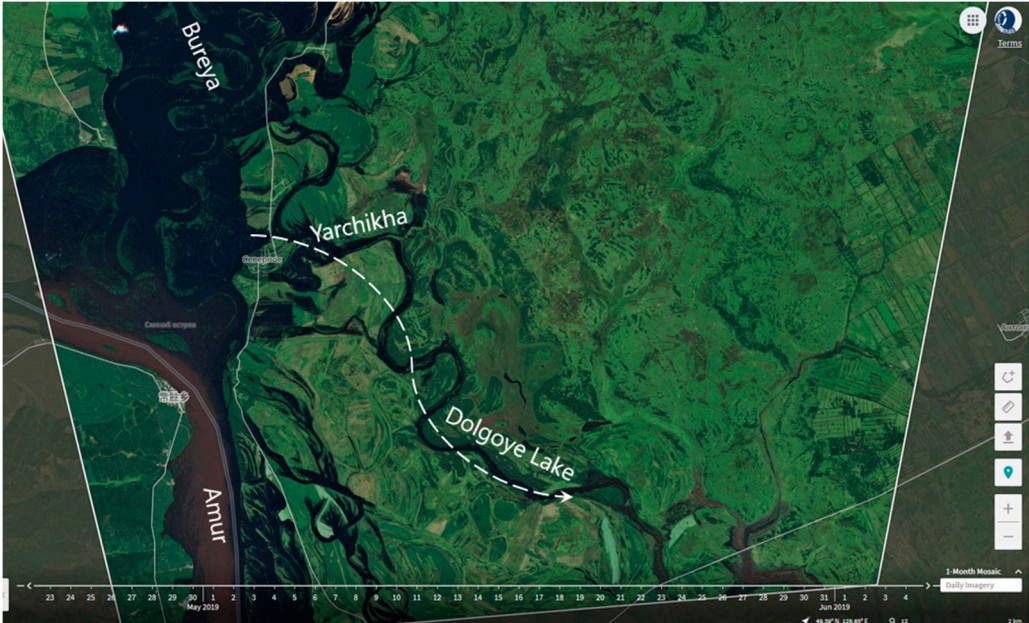

**Figure 6.** On 20 August 2013, the water of the Bureya River entered the Dolgoye Lake through the Yarchikha River, ensuring the hydrological connectivity of the Bureya River and its floodplain wetlands.

A combination of the following conditions led to bank overflow and water access to the floodplain lakes:

- a high-water level of the Amur River (800 cm above the reference point of the Innokentyevka stream gauge) throughout August;
- excessive amount of precipitation (324 mm in August with a climate average of 143 mm for the Arkhara meteorological station); and
- the water release from the Bureya reservoir of 3700 m$^3$/s, which provided the water levels of Bureya River at the Malinovka stream gauge of about 400 cm.

However, these conditions did not lead to the washing out of the lakes within the Bureya floodplain in the Antonovskoye Forestry (see Figures 1 and 7A). At the same time, high water levels of the Amur River, during the 2013 flood, washed out the lakes in the Amur floodplain in the Lebedinskoye Forestry, located downstream from the mouth of the Bureya River, as this area was less affected by flow regulation (see Figure 7B).

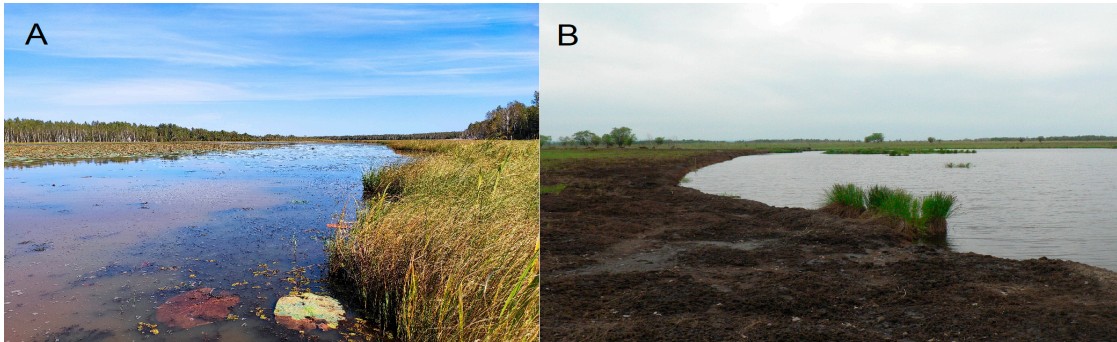

**Figure 7.** (**A**) Dolgoye Lake in the Bureya floodplain was not washed out by the 2013 flood, by O. Nikitina; (**B**) Lebedinoye Lake in the Amur floodplain was washed out by the 2013 flood of overgrowth, by T. Parilova.

In 2019, high water levels of more than 400 cm above the reference point of the Malinovka stream gauge were observed on the Bureya River, while the Amur River levels exceeded 800 cm above its reference point at Innokentyevka. The combination of the Bureya and Amur rivers' high-water levels along with abundant rainfall (225 mm in July as compared with the climate average of 130 mm) ensured the overbank flow and the floodplain inundation, as well as the water inflow into the floodplain lakes. The maximum discharges ranged from 5160 to 5600 m$^3$/s [81]; the lakes were not washed out due to the decrease in the maximum levels by the dam and the extension of the flood peak in time (see Figure 8).

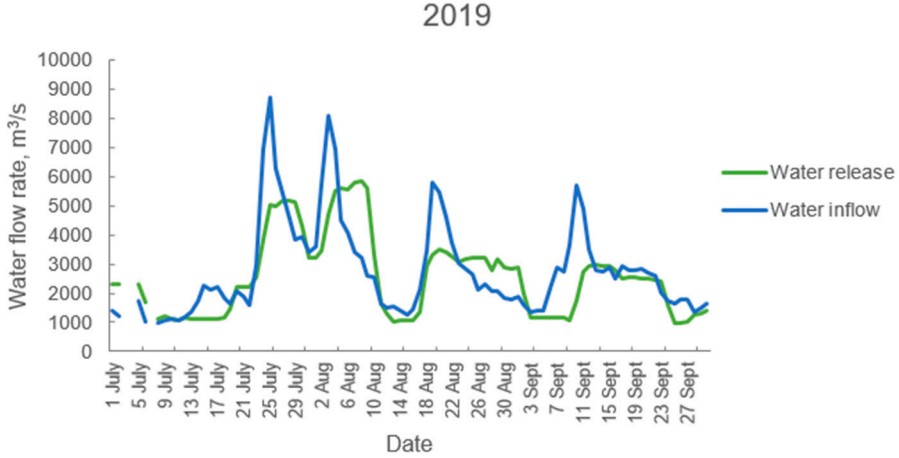

**Figure 8.** Water inflow to and releases from the Bureya reservoir during high-water period of 2019.

Comparison of hydrographs of 2013 and 2019 (Figures 5 and 8) reveals that, despite the higher flow rates of water released from the Bureya reservoir in 2019, the flood of 2013 was more severe because the water level of the Amur River during a major high-water period was higher than in 2019 (see Figure 9). This confirms the importance of the high-water level of the Amur River for defining the flooding and environmental flow conditions in the Lower Bureya River basin.

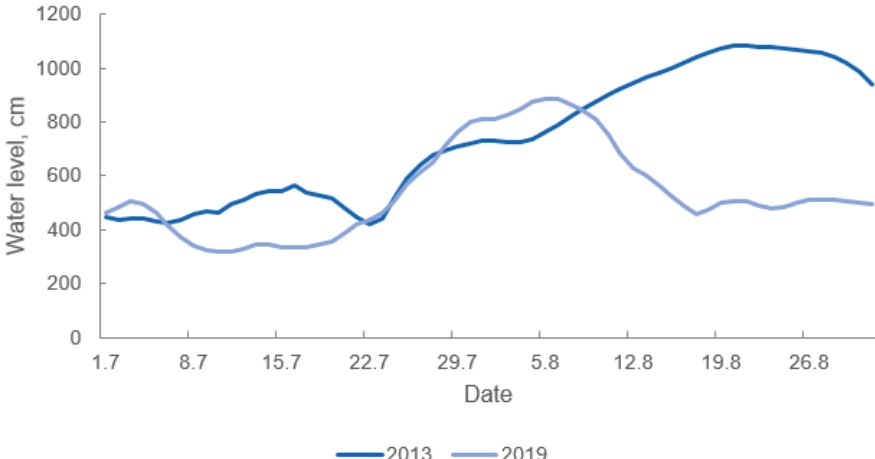

**Figure 9.** Water level of the Amur River at Innokentyevka stream gauge used to be higher for a major high-water period in 2013 as comparing with in 2019.

## 4. Discussion

Next, we interpret our findings in order to develop recommendations for reducing the negative impact of flow regulation and protecting freshwater and floodplain ecosystems in the Amur River basin.

### 4.1. Environmental Flow as a Basis for Freshwater Ecosystem Conservation

The construction of large dams, along with other negative factors, has led to the transformation of many rivers in Russia, including the Amur River, and their freshwater and floodplain ecosystems. For the Amur River basin, dam operation and flow regulation have led to the following: intra-annual flow redistribution and changes to fluvial processes, disturbance in the conditions for natural reproduction of aquatic organisms, a decrease in the area and period of floodplain flooding, loss of hydraulic connection between the river and its floodplain, habitat fragmentation and transformation, changes in species composition and diversity loss, a loss of fish spawning grounds and a decrease in fish catches, etc. [11,12,82].

Water management planning should consider environmental flow as a basis for preserving and restoring freshwater ecosystems. The recently released Emergency Recovery Plan for Freshwater Biodiversity promotes the accelerated implementation of environmental flows as a priority action to conserve and restore freshwater biodiversity. Other priority actions should also consider improving water quality, protecting and restoring critical habitats, sustainably managing the exploitation of freshwater species and river aggregates, preventing and controlling non-native species invasions, and safeguarding and restoring freshwater ecosystem connectivity, including the protection of the remaining free flowing rivers [13]. All these measures are highly relevant to the Amur River basin and should be further developed.

### 4.2. Interpretation of the Results of the Environmental Flow Assessment in the Zeya River Basin

For the Zeya River basin, the problem of exceeding the determined values of water withdrawal is not relevant currently and for the foreseeable future, although further assessments of critical conditions and water withdrawal are important for small rivers of the Middle Amur River basin actively being used for agriculture [83–85].

However, dam regulation of the intra-annual flow redistribution of the Zeya River, leads to water withdrawal during the warm season, which is important for reproduction of aquatic biological resources, as well as floodplain inundations. Under natural conditions, during the May to September warm season, the Zeya River flow at the Belogorye stream gauge has contributed up to 95% of the annual volume [18,45,46]. Due to flow regulation, the flow proportion in the Zeya River lower reaches, for the warm season, is 65–75% of the annual flow [45,46], which is 15–20% lower than

the calculated environmental flow (see Table 1) and indicates a high level of Zeya River freshwater ecosystem transformation.

To ensure sustainable spawning of phytophilic fish and to maintain the floodplain ecosystems, the overbank flow should occur at the Lower Zeya reaches in wet years (25% probability of exceedance), at least once every four to five years. The average lifespan of the Amur pike (*Esox reichertii*) is three to five years, the Prussian carp (*Carassius gibelio*) lifespan lasts six to seven years, and the lifespan of the Amur carp (*Cyprinus carpio*) is more than 12 years [20]. In wet years, the environmental flow volume of high-water years should reach 48 km$^3$ during the May to September period (see Table 1). According to the natural flow regime, the Zeya River discharge at the Belogorye stream gauge should exceed 6500 m$^3$/s for 15–20 days in June and July and be formed by the releases from the Zeya reservoir and lateral inflow. Environmental flow release indicators for the Zeya basin could be fish stocks and catches under different water conditions, and the meadow vegetation productivity of the floodplain.

The analysis shows that bank overflow has occurred only once every five to ten years and for less than 10 days since the Zeya dam was fully completed in 1984. This is not enough to ensure proper conditions for the reproduction of aquatic organisms. In addition, environmental and fishing flow releases at the Zeya dam are unfeasible for technical reasons, as the opening of the spillway gates of the dam can only be carried out after the reservoir is filled to the 317.5 m mark. During the winter period, the reservoir level decreases to 309–310 m, and during the summer flood period, the level rises to 313–315 m, rarely reaching 317.5 m [86].

Changing the operating rules of the existing dams is important to restore freshwater and floodplain ecosystems and ecosystem services that have already been affected by the dam construction and operation. However, if the environmental flow releases were not initially included in early project stages, it may become technically impossible during operation, as evidenced by the Zeya hydropower dam. Furthermore, long-term dam operation reduces the likelihood of environmental flow releases being implemented since floodplains have already been developed by humans and floods and would have resulted in socioeconomic losses. Therefore, any construction within the floodplain that affects the state of the ecosystem should be strictly limited and thoroughly examined, especially within the area of a low floodplain, which should be noted in the Water Code.

In the future, technical restrictions for water releases from the Zeya reservoir below the 317.5 m mark should be solved, in order to allow environmental flow release implementation. For example, in 2011, an additional shore spillway of the Sayano-Shushenskaya hydropower dam was commissioned on the Yenisey River. The Sayano-Shushenskaya dam is the largest hydropower station in Russia in terms of installed capacity. The spillway was built, because of the need to improve the reliability and safety of hydraulic structures, to allow an additional passage of flow rates up to 4000 m$^3$/s during high-water periods and floods, and thereby reduce the load on the dam body [87]. Creation of a similar technical solution at the Zeya dam would allow implementation of environmental flow releases and also improve regulation of the Zeya dam management during severe and catastrophic floods. In turn, this would reduce the negative social and economic consequences of floods.

The large-scale flood of 2013 demonstrated that the floodplains of the Lower Zeya and the Amur rivers below their confluences were inundated, and the oxbow lakes were washed out, with high levels caused by the flow of unregulated tributaries of the Zeya River, such as the Selemdzha, Tom, Urkan, and Dep rivers. The average long-term value of the lateral inflow of the Zeya River, in the section from the Zeya hydropower dam to the Zeya mouth, is about twice as large as the inflow into the Zeya reservoir [35]. With their natural water regimes, the tributaries contribute to the preservation of the most ecologically valuable freshwater ecosystems of the Lower Zeya River basin. They also provide optimization of the thermal regime and other ecological functions [3], thereby improving conditions for reproduction of aquatic organisms in the Lower Zeya River basin. This indicates the importance of their preservation and the need for preventive protection from future flow regulation by the dams.

*4.3. Determination of Environmental Flow Releases From the Bureya Reservoirs and Recommendations for Their Implementation*

The interim operating rules for the Bureya reservoir include the option of implementing environmental flow releases to ensure environmental sustainability of the floodplain area located downstream from the Bureya dam [88,89]. The environmental flow releases should be aimed at periodic inundation of the floodplain, its wetlands, especially the floodplain lakes and channels. The releases should maintain the hydrological connectivity of the rivers and lakes and prevent the lakes overgrowth. The environmental flow release should be defined considering flood frequency and magnitude during the natural flow regime, which would ensure inundation of the floodplain and its wetlands and washing of the floodplain lakes.

The Bureya reservoir and the Bureya lower reaches have no commercial fishery value and the natural habitat for fish has been significantly disturbed by flow regulation [63]. Commercial fishing demands have been fulfilled by the construction of the Aniuysky fish hatchery in the Khabarovsky Krai as a compensation measure for the Bureya dam construction and operation. As agreed with the fishery authorities, releases from the Bureya reservoir to maintain conditions for fish should not be provided [77]. Therefore, the commercial fishery requirements were not considered when defining environmental flow releases.

Under flow regulation, wetlands inundation of the Bureya floodplain occurs under a combination of the following factors:

- a high-water level of the Amur River of more than 800 cm above the reference point at the Innokentyevka stream gauge;
- a water level of the Bureya River of 400 cm or more above the reference point at the Malinovka stream gauge, with water discharge from the Bureya and the Lower Bureya reservoirs of more than 3700 $m^3$/s;
- excessive rainfalls in July and August, providing a large amount of water in the floodplain lakes.

If the Amur River water level is not high, the Bureya River bank overflow occurs at a discharge rate of 6000 $m^3$/s at the Malinovka stream gauge. The bank overflows maintain the hydrological connectivity of the Bureya River with the valuable wetlands of its floodplain. To ensure sustainable conditions of the freshwater ecosystem, the environmental flow release should also wash out the oxbow lakes at a flow rate of 7000 $m^3$/s, while ensuring the conditions for non-flooding of settlements [90,91].

On the basis of the obtained results, the following recommendations for the environmental flow release implementation from reservoirs on the Bureya River are proposed:

1. To ensure the hydrological connectivity of the Bureya River and the wetlands of its floodplain in the lower reaches of the Bureya River, the water should be released at rates of 3700 to 7000 $m^3$/s for 10–15 days in July or August. The release rate should consider the Amur River level at the Innokentyevka stream gauge to ensure settlements are not flooded while providing the environmental flow for ecosystem conservation.
2. To halt the lakes overgrowth by cleaning the lakes of macrophytes, the duration of the maximum water releases of 6000–7000 $m^3$/s should last for two to three days in July or August. The value of the flushing flow rate can be adjusted during hydrological monitoring.
3. Environmental releases should be implemented at least once every six to seven years. This frequency exceeds the frequency of large floods under natural water conditions, which have occurred once every 10–11 years. Given the reduction in flood magnitude, a reduction in the time span is a compensation measure aimed at preserving wetlands under flow regulation.

Environmental flows should be released from the Bureya reservoir because it retains a large volume of water with sufficient capacity to provide such high volumes, unlike the smaller Lower Bureya reservoir to which it is coupled.

Indicators of the effectiveness of environmental flow releases could include the area of lakes and the rate of their overgrowth, the washing of channels and floodplain lakes from organic debris, the food supply for birds, i.e., the abundance of fish inhabiting wetlands, and the number of ichthyophagous birds, including storks and cranes.

The effectiveness of environmental flow releases can be assessed by hydrological monitoring. In 2019, we organized hydrological monitoring on the lakes of the Khingansky Nature Reserve with the support of the World Wide Fund for Nature in order to assess lake dynamics under the influence of flow regulation and climate changes. Subsequent expansion of the monitoring program should allow us to determine the water levels at which the water of the Bureya side enters the Dolgoye Lake through the Yarchikha River. Field observations could be supplemented with satellite imagery or terrain surveys using drones to identify the conditions under which the Bureya bank overflow occurs and hydrological connectivity is ensured.

Environmental flow release implementation in the Bureya River basin should help to preserve the valuable wetlands of the East Asian–Australasian Flyway, and therefore contribute to conservation of endangered bird species. In particular, the environmental releases should help to reduce the negative effect of flood regulation during droughts and to preserve the nesting grounds of cranes. In addition, these releases should indirectly contribute to the improvement of the conditions of natural fish reproduction in the Amur River, downstream of the confluence of the Bureya River, due to the spawning grounds inundation.

Furthermore, the filling of the Bureya reservoir in July and August should be limited by the reservoir to a level of 254.0 m in order to avoid the inundation of the Chekunda settlement located upstream from the dam [90]. This point is an additional incentive for environmental flow implementation. An additional effect of environmental flows would be the floodplain preservation of the Lower Bureya and the Amur rivers below the confluence of the Bureya River from anthropogenic development, primarily from agriculture and settlement development. Along with minimizing the negative impact of flow regulation on freshwater and floodplain ecosystems, this would contribute to the Amur River basin adaptation to floods.

In order to increase the effect of the environmental flows and to improve hydrological connectivity of the Bureya River and the floodplain lakes, it is necessary to expand culverts to allow the Yarchikha's water to easily flow under a road to the Dolgoye Lake in the Khingansky Nature Reserve.

The recommendations developed will be presented to the Federal Water Resources Agency for further discussion and clarification, in order to be included in the Operating Rules for reservoirs of the Bureya River and further implementation.

A relevant example of an environmental flow release is a dam re-operation of the Three Gorges Dam on the Yangtze River in China. The main functions of the dam include producing electricity and controlling floods downstream from the dam. Since 2011, environmental flow releases have been promoting Chinese carp spawning and propagation. However, although the environmental flow releases have improved environmental conditions, the number of fish is still far below the baseline values before the dam construction [17,88]. Another encouraging example is a water reservation for environmental purposes in the San Pedro Mezquital River Basin in Mexico. The wetlands of the large free-flowing river basin include mangrove forests belonging to the Marismas Nacionales Biosphere Reserve, which have been recognized as a Ramsar Site. The river is not water stressed in its middle and lower reaches, however, there have been concerns about future possible development in the river basin, including the Las Cruces hydropower dam construction. In 2014, an Environmental Water Reserve was established, with approximately 80% of its mean annual flow aimed at ensuring water and nutrient supply to the valuable wetlands, while the biological monitoring was focused on their mangroves [17,89].

### 4.4. The Legislative Changes Needed

Global experience demonstrates that the key factors for successful implementation of environmental flows include legislation on environmental flow and research on the impact of dams on the environment, as well as experimental monitoring of the efficiency of the environmental flow implementation [17,92]. For example, environmental flows have been incorporated into water legislation in South Africa and implemented through legally mandated catchment management agencies. The Crocodile River is an example of an environmental flow implementation. Another example of proactive reservation of flows for the environment is the National Water Reserves Program in Mexico [17,89]. This initiative sets sustainable water allocation limits for 189 rivers across the country, considering the social and economic benefits of environmental flows. The current water legislation in Russia does not include the requirements for environmental flows. The definition of terms for permitted surface water withdrawal, environmental flow, and environmental flow release should be introduced into the Water Code of the Russian Federation to indicate their key role in the freshwater ecosystem conservation.

### 4.5. Impacts of Climate Change on Flow Regime and Adaptation to Floods

Global climate changes have impacted the hydrological regime of the rivers in the Amur River basin [93], causing an increase in the frequency and power of floods, erosion of riverbanks, and instability of ice phenomena in the rivers [94]. Over time, climate-driven transformation of freshwater ecosystems can lead to a change in the habitats of flora and fauna [1,95]. Climate change and its associated risks should be assessed to develop appropriate adaptation measures for a river basin [33,94].

The studies conducted for the Zeya and Bureya rivers confirm that the maximum flows have an important ecosystem role in the maintenance of freshwater and floodplain ecosystems and their ecosystem services in the Amur River basin. At the same time, floods cause socioeconomic losses for the Amur region. Therefore, when determining the environmental flow regime, the allowable maximum flow reduction should be determined, as well as the magnitude, timing, frequency, and duration of maximum flows. A sufficient value for the maximum flow should both ensure the ecosystem functions of water and floodplain ecosystems, as well as consider the socioeconomic aspects, including the protection of territories from catastrophic floods. Further assessments of environmental flows should consider various climate change scenarios and their impacts on flow regimes.

Infrastructure development in floodplains has increased the number of people affected by floods in the Amur River basin. Following the results of the catastrophic flood of 2013, the recommendations for flood management have primarily included hard engineering measures such as construction of large dams and reservoirs, levee systems construction, and canal widening and deepening. Some reservoirs have been proposed on already modified tributaries such as the Zeya River, while other designs have targeted still free-flowing tributaries, such as the Selemdzha River [41,42,96,97]. In our opinion, construction of additional flood protection dams and reservoirs will not solve the problem of catastrophic floods. At the same time, the creation of reservoirs has been associated with a significant impact on ecosystems [3–7,9,11,12,72,82]. Therefore, the number of new dams should be strictly limited. Prior to the flood control reservoir project design, less harmful alternatives to the environment should be evaluated and carefully analyzed within the basin management plans. In the case of an urgent need to construct a dam, environmental flow releases should be included at the earliest stages of the project, taking into account basin planning and assessing the impact of a dam on the entire river basin instead of assessing the impact on the local area.

A broader approach to flood adaptation should emphasize the need to implement nature-based solutions, such as protection of flood retention capacities of the floodplains and their wetlands [33,96,97]. Additionally, improved land management at the watershed level is needed to find the optimal balance between the ecosystems' ability for water retention and regulation and economic development. These solutions can reduce the impact of extreme floods, while also helping to sustain floodplain ecosystems which are among the most productive and biodiverse habitats on the planet.

Economic, social, and environmental costs of floodplains properly managed for flood retention are lower than the costs of flood-retention reservoirs and other hard engineered measures. During the 2013 flood, in the Amur River basin, the volume of water accumulated by floodplains was higher than the volume of the existing and any planned hydropower reservoirs together [97,98]. In the future, recommendations should be prepared for establishing protected areas in floodplains that are particularly important in terms of accumulating floodwaters and preserving valuable natural ecosystems.

## 5. Conclusions

Floods are important for maintaining the biodiversity of freshwater and floodplain ecosystems in the Amur River basin. During high floods, the Amur River overflows for tens of kilometers and forms a backwater with its tributaries, inundating their floodplain ecosystems, washing out old lakes, providing a positive effect on fish reproduction, and increasing the fertility of floodplain soils. At the same time, floods cause socioeconomic losses for the Amur region. On the Zeya and Bureya rivers in Russia, three large dams have been built to generate electricity and protect the population from floods.

Flow regulation is one of the key factors determining the state of freshwater ecosystems in the Amur River basin. Decreased maximum water levels downstream of the Zeya dam have led to a reduction in fish spawning grounds and a decrease in their food supply, which have severely reduced fish numbers. Floodplain lakes and river channels have become overgrown. Rare inundation has caused changes in the land use of the floodplain areas of the Zeya River. Flow regulation has worsened the habitat for birds affected by dams in the Bureya floodplain, which has been especially important for the Ramsar site of the Khingano-Arkharinskaya Lowland, an important nesting area for endangered bird species such as the Oriental stork (*Ciconia boyciana*), red-crowned crane (*Grus japonensis*), and white-naped crane (*Grus vipio*).

To reduce the negative impact of flow regulation and to preserve freshwater and floodplain ecosystems, it is necessary to implement environmental flow releases, which would flush the channels, the spawning grounds, and the oxbow lakes; increase the fertility of floodplain soils; and ensure the proper water regime for wetlands, thereby preserving them.

The volumes of permitted surface water withdrawal and environmental flow for different water years and seasons have been determined for the entire Zeya River basin. The problem of surface water withdrawal is not relevant for the Zeya River basin, since the actual water withdrawal is less than 1% of the permitted volume. At the same time, the flow proportion during the warm season has decreased from 95 to 75%, which was 10–15% less than the calculated volume for environmental flow and has negatively impacted the state of freshwater water and floodplain ecosystems. It is recommended to inundate the floodplain at least once every five years for 15–20 days. However, environmental flow releases from the Zeya reservoir are impracticable due to technical reasons. Therefore, the importance of preserving the free-flowing tributaries of the Zeya River increases. With their natural water and thermal regimes, the tributaries contribute to the preservation of freshwater and floodplain ecosystems in the lower reaches of the Zeya basin. Future technical improvements at the Zeya hydropower dam would enable implementation of environmental flow releases, as well as improve the Zeya dam management regulation during severe and catastrophic floods and reduce negative social and economic consequences of floods.

To ensure hydrological connectivity between the river and wetlands and washing out of old lakes and channels in the Bureya floodplain, we recommend providing water discharges of 3700 to 7000 m$^3$/s from the Bureya reservoirs for 10–15 days in August. To flush the oxbow lakes, the high-water discharges of 6000–7000 m$^3$/s from the Bureya reservoirs should last for two to three days. Environmental flow releases should be implemented at least once every six to seven years. They should help preserve the important Ramsar wetlands of the Khingano-Arkharinskaya Lowland, which provide habitats for populations of endangered bird species while avoiding flooding of settlements. In addition, environmental flow releases should indirectly contribute to improved conditions for natural reproduction of fish in the Amur River, downstream from the confluence of the Bureya River.

An additional effect of the environmental flow releases would be the preservation of the Lower Bureya floodplain from development within flood-prone areas, which will contribute to the adaptation of the Amur River basin to floods.

Environmental flow legislation is the key factor for successful implementation of environmental flows. The definition of the terms for permitted surface water withdrawal, environmental flow, and environmental flow release should be introduced into the Water Code of the Russian Federation, indicating their key role in freshwater ecosystem conservation. Any construction within the floodplain that affects the state of the ecosystem should be strictly limited and thoroughly examined, especially within the area of a low floodplain, which should be noted in the Water Code.

The conducted studies confirm that maximum flows have an important ecosystem role in maintaining freshwater and floodplain ecosystems and their ecosystem services in the Amur River basin, but, at the same time, cause socioeconomic losses for the population. Therefore, when determining environmental flows, the allowable maximum flow reduction should be determined. A sufficient flow value should ensure the ecosystem functions of water and floodplain ecosystems while considering the protection of territories from catastrophic floods. Further environmental flow assessments should consider scenarios of climate change and their impact on the flow regime.

Global climate changes can lead to an increase in the frequency and power of floods. Prior to flood control reservoir project design, alternatives that are less harmful to the environment should be evaluated. Nature-based flood adaptation to floods should include measures such as protecting flood retention capacities of the floodplains and establishing protected areas in floodplains for simultaneously accumulating floodwaters and preserving ecosystems.

**Author Contributions:** Conceptualization, O.I.N., V.G.D., M.P.P., T.A.P. and M.V.B.; methodology, V.G.D.; validation, O.I.N., V.G.D., M.V.B., M.P.P. and T.A.P.; formal analysis, O.I.N.; investigation, O.I.N., V.G.D., M.P.P. and T.A.P.; resources, O.I.N., V.G.D., M.P.P., T.A.P. and M.V.B.; data curation, O.I.N.; writing—original draft preparation, O.I.N.; writing—review and editing, O.I.N., V.G.D., M.P.P. and M.V.B.; visualization, O.I.N.; supervision, V.G.D.; project administration, O.I.N.; funding acquisition, O.I.N. and V.G.D. All authors have read and agreed to the published version of the manuscript.

**Funding:** This research was funded by the World Wide Fund for Nature (WWF-Russia), grant number WWF001462.

**Acknowledgments:** We wish to thank our peer reviewers for their insightful comments and suggestions. We particularly thank Peter Osipov and Anna Serdyuk (Amur branch of WWF-Russia) for all their help and support throughout this research work. We would like to express our great appreciation to Oxana Erina (Moscow State University) for her enthusiastic encouragement and suggestions during the development of this research work. We are grateful to Yury Darman (advisor of the Amur branch of WWF-Russia) and Eugene Simonov (the Rivers without Boundaries Coalition) for their support and advice on the issue of environmental flows and freshwater ecosystem conservation.

**Conflicts of Interest:** The authors declare no conflict of interest.

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
