# Peer review of "Environmental Flow Releases for Wetland Biodiversity Conservation in the Amur River Basin"

_water, doi:10.3390/w12102812_

Round 1

Reviewer 1 Report

I have reviewed Manuscript Number: ID: water-933331 titled "Environmental flow releases for wetland biodiversity conservation in the Amur River basin". The material is presented appropriately and clearly, the data contained in table and figures represent understandable documentation of the research problem. I do not propose any additional comments. I recommend its publication in the present form.

Author Response

We thank the reviewer for revising our paper and sharing positive feedback.

We hope the reviewer will share the vision on the updated version of the article.  

Reviewer 2 Report

I have just read the manuscript number 933331, entitled "Environmental flow releases for wetland biodiversity conservation in the Amur River basin" and written by Nikitina et alii.

In general I believe that the work is interesting and well organised, although unfortunately I cannot recommend accepting it for publication in Water in its present form but only after some revisions, starting with the language, which is not used homogeneously throughout the text. In fact, English seems very scholastic in different parts of the text, even if in some sections the text is very tight. I therefore recommend asking for help from a native speaker who can make reading more fluent.

Furthermore, the main criticisms concern Abstract, Introduction and Discussion.

Abstract ought to be totally rewritten; Authors have to report own findings avoiding to write general sentences on the utility of this kind of study. 

Introduction can be widely reduced (50%) without losing its meaning; some contents and ideas seem to be repeated several times with different words, overall the concept on the importance to apply right prevention actions.

As for Discussion, I think it is important to propose more comparisons with literature since your (very interesting) findings regard only a small study area, and this let your job suffer a little bit. In addition, it could be good adding some suggestions on how to act for preventing flood events, since “Either you bring at least one solution, or you too are part of the problem” (Confucius).

Please, do not numerize your conclusions.

Additionally, I would take the biological aspects into account. In particular, as far as the aims are concerned, it is necessary to better describe in Introduction and Discussion what are all possible issues concerning habitats and (plant and animal) species, since there could be protected habitats and biota needing such events, explaining if your findings take all or a part of those issues into account. I understand your aims are to evaluate the efficacy to prevent such events, but overall to date sustainable activities ought to be perform to preserve all natural (living and non) resources.

Author Response

We are very grateful to the reviewer who took time to share useful suggestions and remarks.  

Reviewer 3 Report

Environmental flow releases for wetland biodiversity 2 conservation in the Amur River basin

By Nikitina et al.

Introduction

The introduction requires an introductory paragraph – perhaps on river regulation, the world’s largest rivers, freshwater diversity, climate change etc…..

Line 34 – change 3.5 thousand to ~3,500

Line 39 – change to Three hundred and twenty

Line 42 – should it be “providing ecological integrity”?

Line 50 – I am not sure what “a prevailing predominance of rainfall” means. Please reword

Line 53 – when is the ‘warm season’?

Line 70 – the most considerable losses are typical in the agriculture, industrial and communal sectors.

Line 79 – the human? Population

Line 124 – ‘their frequency also sharply declined’ – please reword

Results – suggest combining this with the discussion as most of the results are discussed in detail in the results section, rather than in the discussion. The results also contain recommendations, which should be in the discussion or conclusions.

There is currently a great deal of repetition throughout the manuscript. I suggest shortening the methods and results by removing discussion points. Much of this is also included in the Introduction.

The discussion requires an opening paragraph as to the impacts of river regulation; much of which is evident in this study.

Author Response

We thank the reviewer for a thoughtful revision of our manuscript, valuable observations and suggestions.

Round 2

Reviewer 2 Report

I have appreciated very much the Authors' efforts in modifying the manuscript. Although the reading si more fluent than that of the previous version, I am continuing to observe some small changes that can be made on the English language style, but this is a decision I leave to Editor.

Author Response

Thank you for your comment. We appreciate your opinion, and we are glad to know that you observe our efforts in changing the text according to your comments and suggestions.
Regarding the English language, we asked for help from a native speaker when working on the text. Therefore, we hope that the text became more comfortable to read. We'll wait for the final decision of the Editor.

Reviewer 3 Report

Please note that there are many glitches on the pdf - I have highlighted them in yellow.

The authors have addressed my comments, but it looks like something has gone wrong in the editing.

Author Response

We are glad to hear that you agree with the changes we made addressing your comments.

Thank you for your attention to the text editing. Indeed, there must be a technical mistake, as the text you have seen used to combine both the edits and the previous version. Please see the updated version in attachment, this time without tracking changes.
